# Effect of Tempeh and Daidzein on Calcium Status, Calcium Transporters, and Bone Metabolism Biomarkers in Ovariectomized Rats

**DOI:** 10.3390/nu16050651

**Published:** 2024-02-26

**Authors:** Iskandar Azmy Harahap, Maciej Kuligowski, Adam Cieslak, Paweł A. Kołodziejski, Joanna Suliburska

**Affiliations:** 1Department of Human Nutrition and Dietetics, Faculty of Food Science and Nutrition, Poznan University of Life Sciences, 60-624 Poznan, Poland; iskandar.harahap@up.poznan.pl; 2Department of Food Technology of Plant Origin, Faculty of Food Science and Nutrition, Poznan University of Life Sciences, 60-624 Poznan, Poland; maciej.kuligowski@up.poznan.pl; 3Department of Animal Nutrition, Faculty of Veterinary Medicine and Animal Science, Poznan University of Life Sciences, 60-637 Poznan, Poland; adam.cieslak@up.poznan.pl; 4Department of Animal Physiology, Biochemistry and Biostructure, Faculty of Veterinary Medicine and Animal Science, Poznan University of Life Sciences, 60-637 Poznan, Poland; pawel.kolodziejski@up.poznan.pl

**Keywords:** isoflavones, calcium, bone metabolism, menopause, osteoporosis

## Abstract

Menopause marks a critical life stage characterized by hormonal changes that significantly impact bone health, leading to a heightened susceptibility to bone fractures. This research seeks to elucidate the impact of daidzein and tempeh on calcium status, calcium transporters, and bone metabolism in an ovariectomized rat model. Forty female Wistar rats, aged 3 months, participated in a two-phase experiment. The initial phase involved inducing a calcium deficit, while the second phase comprised dietary interventions across five groups: Sham (S) and Ovariectomy (O) with a standard diet, O with bisphosphonate (OB), O with pure daidzein (OD), and O with tempeh (OT). Multiple parameters, encompassing calcium levels, calcium transporters, bone histopathology, and serum bone metabolism markers, were evaluated. The findings revealed that the OT group showcased heightened levels of bone turnover markers, such as pyridinoline, C-telopeptide of type I collagen, bone alkaline phosphatase, and procollagen type I N-terminal propeptide, in contrast to S and O groups, with statistical significance (*p* < 0.05). Histopathologically, both the OD and OT groups exhibited effects akin to the OB group, indicating a decrease in the surface area occupied by adipocytes in the femoral bone structure, although statistically non-equivalent, supporting the directionally similar trends. Although TRPV5 and TRPV6 mRNA expression levels in the jejunum and duodenum did not display statistically significant differences (*p* > 0.05), the OD and OT groups exhibited increased expression compared to the O group. We hypothesized that obtained results may be related to the effect of isoflavones on estrogen pathways because of their structurally similar to endogenous estrogen and weak estrogenic properties. In conclusion, the daily consumption of pure daidzein and tempeh could potentially improve and reinstate calcium status, calcium transport, and bone metabolism in ovariectomized rats. Additionally, isoflavone products demonstrate effects similar to bisphosphonate drugs on these parameters in ovariectomized rats.

## 1. Introduction

Menopause signifies a crucial life stage for women, often marked by a decline in estrogen hormone levels, leading to various health challenges [1]. Among these challenges, osteoporosis emerges as a major concern. The decrease in estrogen levels during menopause contributes to calcium deficiency, ultimately elevating the risk of bone fractures. Women, particularly those in postmenopausal stages, are highly vulnerable to the detrimental effects of osteoporosis, underscoring the necessity for effective interventions to uphold bone health [2].

While addressing osteoporosis is paramount, the current treatments, such as bisphosphonates, come with notable drawbacks. Despite being widely used for managing bone fractures, bisphosphonates are associated with adverse effects, prompting the exploration of alternative, safe, and long-term solutions. Discovering innovative approaches to improve bone health and prevent fractures is crucial, especially considering the limitations of existing therapeutic options [3]. In light of ongoing debates and controversies surrounding bone health and menopause management [4], it is imperative to explore alternative treatment modalities that address the multifaceted aspects of osteoporosis while minimizing adverse effects.

Bone metabolism constitutes a dynamic process crucial for maintaining skeletal integrity throughout an individual’s life [5,6,7,8,9]. Two primary facets governing bone metabolism are bone resorption and bone formation [10]. Bone resorption involves the breakdown of bone tissue, with key markers of this process including pyridinoline, deoxypyridinoline, and C-telopeptide of type I collagen. Conversely, bone formation is characterized by the synthesis of new bone tissue, with notable markers being bone alkaline phosphatase, osteocalcin, and procollagen type I N-terminal propeptide [10]. The delicate equilibrium between resorption and formation is vital for overall bone health [11,12].

At the core of bone metabolism lies the intricate regulation of calcium homeostasis. Calcium plays a pivotal role in bone mineralization. Within bone tissue, calcium ions (Ca^2+^), mediated by the calcium-sensing receptor, are instrumental in initiating processes like the proliferation of preosteoblasts. Moreover, calcium signaling influences the differentiation of preosteoblasts into mature osteoblasts, along with the synthesis and mineralization of essential bone proteins [13]. Maintaining an optimal balance of calcium levels is crucial for preventing bone disorders and ensuring proper cellular functions [14]. A critical aspect of calcium regulation unfolds in the small intestines, where epithelial calcium transporters, specifically TRPV5 and TRPV6, play a vital role in facilitating Ca^2+^ absorption [15,16,17,18]. Epithelial calcium transporters, such as TRPV5 and TRPV6, are specialized proteins located in the cells lining the small intestine. They are responsible for transporting calcium ions from the intestine into the bloodstream [19,20].

Dysregulation in bone metabolism, especially disruptions in the delicate balance between resorption and formation, can lead to various skeletal disorders, including osteoporosis [11]. Consequently, investigating factors influencing this equilibrium is crucial for developing interventions aimed at preventing or managing bone-related conditions. This includes exploring dietary components and their potential impact on bone health, forming the basis for our current study.

Isoflavones, acknowledged as phytoestrogens, have demonstrated promise in influencing calcium regulation, absorption, and bone metabolism [21]. Phytoestrogens are natural compounds found in plants that have estrogen-like effects in the body. Daidzein and genistein are specific types of isoflavones abundant in soybean. Among the sources of isoflavones, soy tempeh stands out as a noteworthy candidate. The fermentation process amplifies the levels of daidzein and genistein in tempeh, enhancing its nutritional content [22,23,24].

Astawan et al. [25] conducted a 90-day intervention study investigating the impact of various protein sources and levels on calcium absorption and retention, as well as serum and bone calcium content. Interestingly, their findings indicated no significant differences in these parameters across different protein sources and levels in the diet. Similarly, Yoo et al. [26] demonstrated increased bone mineral density and bone mineral content in ovariectomized rats fed fermented soybeans. These results suggest a potential beneficial effect of fermented soybeans on bone health. Furthermore, while our previous study did not utilize an ovariectomized rat model, we found that a diet with daidzein and genistein improves calcium transport in the duodenum and reduces serum concentrations of pyridinoline in healthy female rats [27]. However, it is noteworthy that despite these studies, there remains a lack of detailed discussion on the specific effects of tempeh and daidzein on calcium status, calcium transporters, and bone metabolism markers in ovariectomized rats. The ovariectomized rat model is well-established and widely used for studying menopausal osteoporosis due to its relevance in mimicking postmenopausal bone loss and its advantages in providing controlled experimental conditions. Therefore, our study aims to address this gap by systematically investigating the impacts of tempeh and pure daidzein, contrasting them with a conventional bisphosphonate drug. We hypothesize that tempeh and daidzein intake will lead to improvements in calcium status, calcium transporters, and bone metabolism in the ovariectomized rat model of menopausal osteoporosis. Gaining insights into the influence of isoflavones, especially in the context of soy tempeh, on bone health can offer valuable information for potential dietary interventions in managing menopausal osteoporosis.

This study presents a unique approach by examining the impacts of tempeh and pure daidzein in contrast to the conventional bisphosphonate drug. Employing an ovariectomized rat model of menopausal osteoporosis, our goal is to elucidate the effects of daidzein and tempeh on calcium status, calcium transporters, and bone metabolism. Through this comparative analysis, we aim to contribute to the identification of dietary strategies that could provide safe and effective alternatives for managing bone health in the context of menopausal osteoporosis.

## 2. Materials and Methods

### 2.1. Materials

The study involved 40 female Wistar rats aged 3 months, obtained from the Nencki Institute of Experimental Biology—Polish Academy of Sciences in Warsaw, Poland. The rats were fed the AIN 93M food, purchased from Zoolab in Sędziszów, Poland, as their standard diet. This diet comprised various components, including Augusta variety soybeans obtained from Poznań University of Life Sciences, Poland, pure daidzein procured from Gentaur Molecular Products BVBA in Kampenhout, Belgium, and alendronate sodium trihydrate purchased from TCI Europe N.V. in Zwijndrecht, Belgium.

Additionally, the crucial component, calcium citrate tetrahydrate, was acquired from Warchem Sp. z o.o. in Warsaw, Poland. The strain *Rhizopus oligosporus* NRRL 2710 used in the fermentation process was obtained from the Agricultural Research Service Culture Collection located in Illinois, United States of America. Moreover, we produced tempeh, a fermented soybean product, using *R. oligosporus* NRRL 2710, following the methodology described in our previous investigation [28].

### 2.2. Ethics of Animal Research

Ethical approval was obtained from the Lokalna Komisja Etyczna (Local Ethical Committee) in Poznań, Poland. Registration number 21/2021 was issued on 21 May 2021. The study adhered rigorously to various guidelines and regulations, including the National Institutes of Health’s Handbook for the Care and Use of Laboratory Animals (NIH Publication No. 80-23, Revised 1978), Directive 2010/63/EU of the European Parliament and the Council of 22nd September 2010 on the Protection of Animals Used for Scientific Purposes, and relevant Polish legislation. It is crucial to note that all animal experiments were conducted strictly following the rules established by Poznań University of Life Sciences, Poland, as outlined in the Animal Research: Reporting of In Vivo Experiments (ARRIVE) guidelines.

### 2.3. Adaptation and Conditioning the Animal Lab Environment

The 3-month-old female Wistar rats were housed in the secure and well-regulated conditions of the Animal Laboratory at Poznań University of Life Sciences, Poland. Wistar rats were chosen as the experimental model for this study due to their established sensitivity to estrogen and consistent sexual cycle stability [29]. The rats were accommodated in a chamber maintained at a constant temperature of 21 ± 2 °C, with a relative humidity range of 55–65%, and subjected to a 12-h light/dark cycle. Throughout the adaptation and intervention periods, the rats were paired and housed in stainless steel cages with enamel coating to minimize electromagnetic interference.

A 1-week acclimatization phase was provided for the rats to adjust to the laboratory environment before the commencement of the trial. Labofeed B (Żurawia, Kcynia, Poland) and tap water were freely available to the rats during this phase. In terms of the nutritional requirements for adult rats, Labofeed B adheres to the guidelines established by the National Research Council of Poland.

Stress was minimized by limiting the number of individuals interacting with the animals and the duration of their contact during the trial. Moreover, throughout the experiment, the rats had continuous access to veterinary care. Figure 1 is a research flowchart that visually outlines the research process.

### 2.4. Ovariectomized Operation

After the adaptation period, the rats were divided into three distinct groups: a sham operation group (*n* = 8) and a group that underwent bilateral ovariectomy (*n* = 32). A sham control group has been incorporated into this study to replicate a procedure or treatment experience in a manner that does not involve the actual administration of the procedure or test substance [30]. This group allows for the comparison of outcomes between rats undergoing ovariectomy and those undergoing a sham operation, providing valuable insights into the specific effects of only ovariectomy (without the effect of surgery) on the variables under investigation. All ovariectomy procedures were conducted by highly qualified experts in animal ovariectomy. During the surgical procedures, a mixture of Ketamine and Cepetor was administered for anesthesia induction. Additionally, meticulous attention was given to maintaining stringent cleanliness practices and sterile conditions throughout the surgical interventions. To enhance surgical accessibility, the dorsal region of each rat was depilated, and a linear incision measuring approximately 10–15 mm was made within the designated operative site. Following the surgical procedure, each rat was placed on a heated mat that was carefully regulated to maintain a temperature of roughly 30 ± 1 °C, creating a favorable and soothing atmosphere to facilitate recuperation. Throughout the post-operative period, the rats were closely monitored for any signs of distress, pain, or complications, including changes in behavior, appetite, and mobility. Any observed abnormalities were promptly addressed by the veterinary team to ensure the welfare of the animals and minimize any potential impact on the study outcomes. A period of 7 days was dedicated to the attentive implementation of continuous monitoring to safeguard the welfare of all rats that underwent surgery. After the completion of the surgical operations, the rats were provided with a semi-synthetic meal formulated according to the AIN-93M guidelines [31]. Additionally, they were given free access to tap water throughout this stage.

### 2.5. Grouping the Rats

After a 7-day recovery period, the initial body weights of the rats were measured using a calibrated scale. Implementing this process is crucial to properly randomize the experimental groups, ensuring that each group begins the investigation from a comparable initial state. Subsequently, the rats that underwent ovariectomy were randomly assigned to seven groups, each comprising eight rats, based on their body weight. The inclusion of randomization of animals holds utmost significance in the context of experimental design, as it minimizes the potential for bias and ensures the comparability of groups at the commencement of the study [32]. We acknowledge the importance of ensuring adequate statistical power in our study design. Previous studies have indicated that a minimum of eight rats in each group is sufficient to achieve appropriate statistical power for detecting a significant effect of the intervention [33]. Therefore, we utilized eight rats in each group to ensure the study’s design had adequate power to detect significant differences.

### 2.6. Conditioning the Calcium Deficiency

Group 1 comprised rats assigned to the sham group (S; *n* = 8) and were fed a standard diet with a calcium deficiency. Group 2 consisted of ovariectomized rats (O; *n* = 32) who were subsequently provided with a standard diet exhibiting a calcium deficiency. The calcium-deficient diet was formulated by removing calcium from the mineral composition of the regular diet. Throughout the investigation, the researchers meticulously observed and recorded the rats’ daily dietary consumption over 3 weeks. Additionally, the rats had unrestricted access to deionized water. Based on a prior investigation, it has been determined that 3 weeks is sufficient for the induction of calcium deficiency [34]. Our objective was to establish the impact of daidzein and tempeh intakes on bone health parameters. To achieve this, we commenced the study with a calcium-deficient diet. By employing this methodology, we were able to simulate a practically significant scenario and establish a strong basis for examining the possible therapeutic effectiveness of our interventions in reducing osteoporosis caused by calcium deficiency.

### 2.7. Modified Diet Intervention

After inducing calcium deficiency, both the sham (S) and ovariectomized (O) groups were given a standard diet with calcium citrate tetrahydrate. The ovariectomized (O) group was further divided into four subgroups, each comprising eight rats, and these subgroups received a standard diet enriched with calcium citrate tetrahydrate. Table 1 displays the diet formulas. Throughout the 6-week intervention period, the rats had unrestricted access to the diets and deionized water.

The selection of daidzein and tempeh doses was informed by prior studies that established their beneficial impacts on the skeletal well-being of rodents and humans. Through a comprehensive analysis of aglycon content, specifically daidzein, glycitein, and genistein, it has been demonstrated that the optimal daily consumption of tempeh should be set at 250 g [35]. The dosage of daidzein at 10 mg/kg diet was chosen based on our previous research [27,28,36,37,38], and the quantity of tempeh corresponds to the daidzein content. In this study, the amount of tempeh was adjusted to the appropriate amount of pure daidzein. To incorporate 250 g/kg of tempeh flour and 10 mg/kg of daidzein into the AIN93M diets, we carefully adjusted the diets by replacing starch with 250 g and 10 milligrams of the corresponding ingredients. This approach ensured that the diets had the correct amount of tempeh flour or daidzein while being nutritionally consistent.

We determined the dosages of isoflavones by referencing prior studies that have documented their positive impacts on bone health in both rodents and humans [35]. Furthermore, long-term therapy with alendronate bisphosphonate at a dosage of 3 mg/kg/day has been shown to benefit fracture healing and bone remodeling in ovariectomized rats [39]. The dose of alendronate bisphosphonate was adjusted weekly after measuring the rats’ body weight.

### 2.8. Body Weight and Food Intake Monitoring

The rats in each group underwent weekly weighing using a calibrated scale throughout the intervention phase. Throughout the experiment, researchers meticulously documented the daily diet consumption for each group. To ensure continuous access to sustenance, the rats received a daily supply of fresh food and deionized water, with any remnants from the previous day promptly removed. This standardized procedure guaranteed consistent food and water conditions, crucial for preserving the rats’ well-being and supporting their regular development [40]. Monitoring weight and food intake was deemed crucial, as any alterations in these variables could signal potential health concerns, underscoring the necessity for ensuring the reliability and validity of the experiment. Moreover, the daily provision of fresh food and water, coupled with the removal of any remaining food, effectively prevented spoilage, ensuring that the rats consumed an untainted and fresh diet and water throughout the experiment.

### 2.9. Body Composition Analysis

Three days prior to decapitation, the body composition of rats in each experimental group was assessed using Bruker’s All New 2nd Generation Minispec LF90II Body Composition Analyzer. This analysis enabled the measurement of fat mass (in grams).

### 2.10. Decapitating the Rats

Upon completion of the intervention phase, a fasting period lasting 4–6 h was imposed on the rats, and their body weight was measured on a calibrated scale. This fasting protocol aimed to minimize the potential impact of recent food intake on subsequent weight measurements. Following the weight assessment, euthanasia was performed on the rats through decapitation, and a blood sample was collected for subsequent blood morphology analysis.

### 2.11. Blood, Serum, Bone, and Fecal Collection

After a 6-week dietary intervention, rats underwent a 12-h fasting period. Following the measurement of their body mass, the rats were decapitated, and tissue and blood samples were collected. Whole blood was obtained via cardiac puncture using anticoagulant-treated tubes, while serum samples were collected using serum-separated tubes. Additionally, serum samples were transferred to sterilized tubes, and the blood was allowed to clot at room temperature for 30 min. Subsequently, the samples underwent centrifugation at 4 °C for 15 min at 2000 rpm to separate blood cells. The resulting supernatants were collected and stored at −80 °C.

### 2.12. Blood Morphology and Bone Histopathology Measurements

Whole-blood morphological and bone histopathology measurements were performed in a certified commercial laboratory (Alab Laboratories, Poznań, Poland). The methodology employed in this study followed a rigorous step-by-step process for handling and preparing tissue samples, ensuring their optimal quality and suitability for subsequent histopathological examination. Upon arrival in the laboratory, tissue specimens underwent an initial macroscopic evaluation to assess the degree of fixation. If any tissues were inadequately fixed, they were bisected sagittally and then further fixed to ensure preservation and stability.

Following the fixation assessment, the tissue samples underwent decalcification for 14 h using an ionic decalcifier ethylenediaminetetraacetic acid (EDTA) solution. They were then immersed in 70% ethanol for a minimum of 24 h to complete the fixation process. After fixation and decalcification, the tissues underwent another round of macroscopic evaluation to ascertain the level of preparation and suitability for further histopathological examination. Trimmed sections of the specimens were carefully sealed in appropriately labeled histology cassettes to maintain their integrity and proper identification throughout the process. Histological preparation followed established protocols for standard histological paraffin techniques. The sealed cassettes containing tissue sections were processed in a tissue processor, undergoing a series of alcohol solutions with increasing concentrations, culminating in xylene treatment, all following established histological paraffin techniques. Once embedded in paraffin blocks, the tissue specimens were precisely sectioned using a histological microtome, resulting in thin slices suitable for subsequent staining procedures. For histological staining, standard haematoxylin-eosin topographic staining was employed to reveal cellular and tissue structures.

The methodology utilized for microscopy assessments and consultations in this study was executed with meticulous precision to ensure precise histopathological evaluation and documentation. A team of experienced veterinarian-pathologists conducted the histopathological assessments using Axiolab 5 microscopes from Zeiss in Halle, Germany. The criteria for histopathological evaluation were derived from a thorough review of scientific literature and recommendations provided by The Global Editorial and Steering Committee (GESC) in their International Harmonization of Nomenclature and Diagnostic Criteria (INHAND) guidelines.

The evaluation of tissue specimens was conducted at various magnifications, specifically 5×, 10×, and 40× objective magnification, enabling pathologists to thoroughly scrutinize the location, nature, and severity of pathological changes within the samples. To standardize the grading assessment, a catalog of histopathological changes was established, drawing upon literature, team expertise, and the initial review of preparations. Each identified change underwent evaluation on a scale ranging from 0 (none) to 4 (severe), providing a quantitative measure of the extent of pathology. Additionally, three distinct B scales were employed to assess the severity of each specific lesion.

In the morphometry aspect of the study, measurements of the surface area and width of trabeculae within spongy bone were undertaken. This process was conducted in two compartments: the spongy bone of the distal epiphysis below the epiphyseal line and the spongy bone of the distal epiphysis above the epiphyseal line, situated within the medullary area. Three random areas within each compartment were chosen for assessment at a 10× high-power field, with a total area of 1,687,500 µm^2^ per compartment. Trabecular surface area measurements were recorded and subsequently presented as both total and relative areas (percentage). Additionally, the width of trabeculae was extensively measured, encompassing 80–120 measurements per compartment, ensuring a comprehensive understanding of trabecular structure.

To document the microscopic findings, images of the most representative areas in all study groups were captured using a 3DHISTECH PANNORAMIC 250 Flash III microscope with a 10× objective. These images were saved in JPG format with a resolution of 3840 × 2160 pixels, and a scale was incorporated for reference. Furthermore, all slides were digitized as Whole Slide Images (WSI) through the Grundium Ocus^®^20 microscope slide scanner, stored in the SVS Aperio format, comprising single-file pyramidal tiled TIFF images with nonstandard metadata and compression. This comprehensive methodology ensured precise histopathological assessment and meticulous documentation of the findings through high-quality photographic and digital records.

### 2.13. Calcium Content Measurement

The calcium concentration in the diet, serum, bone, and fecal samples was determined using flame atomic absorption spectrometry (AAS-3, Carl Zeiss, Jena, Germany) after appropriate dilution with deionized water and 0.5% Lanthanum (III) chloride (Merck KGaA, Darmstadt, Germany). The calcium content in these samples was quantified at a specific wavelength of 422.7 nm. To assess the precision and reliability of this analytical technique, we employed a certified reference material, specifically Bovine Liver 1577C (Sigma-Aldrich, St. Louis, MO, USA). The results obtained from the analysis of this reference material demonstrated a notably high level of method accuracy, with a calculated accuracy rate of 92% for calcium quantification.

### 2.14. Bone Biomarkers Measurement

We utilized commercial enzyme-linked immunosorbent assay (ELISA) kits obtained from Qayee Bio-Technology Co., Ltd., Shanghai, China, in conjunction with absorption spectrophotometry (LEDetect96, Labexim, Lengau, Austria) to quantify serum levels of markers associated with bone metabolism. Specifically, pyridinoline, deoxypyridinoline, and C-telopeptide of type I collagen were measured as biomarkers of bone resorption, while bone alkaline phosphatase, osteocalcin, and procollagen type I N-terminal propeptide were measured as biomarkers of bone formation.

### 2.15. Calcium Transporters Analysis

The presence of calcium transporters was assessed through quantitative real-time polymerase chain reaction (RT-PCR). Total RNA extraction from duodenum and jejunum tissues was performed using EXTRAzol (Cytogen, Zgierz, Poland). Subsequently, EXTRAzol was introduced into separate PCR tubes, and mechanical homogenization of the sample material was achieved using TissueLyser II (Qiagen, Germantown, MD, USA). The High-Capacity cDNA Reverse Transcription Kit (Life Technologies, Grand Island, NY, USA) was employed to reverse-transcribe 1 μg of total RNA into cDNA. RT-PCR of the collected DNA was carried out on QuantStudio 12K Flex™ using gene-specific primers and 5× HOT FIREPol^®^ Eva-Green^®^ qPCR Mix Plus (ROX). Melting points of the DNA were measured (transition rate of 0.1 C/s) to assess specificity. Relative gene expression was analyzed using the ^ΔΔ^CT method, with Gapdh serving as a standard. TRPV5 and TRPV6 mRNA levels were expressed as arbitrary units relative to Gapdh mRNA levels. The complete list of PCR primers is provided in Table 2.

### 2.16. Statistical Analysis

Statistical significance for identified differences, as determined through analysis of variance, was evaluated using Tukey’s post hoc test for multiple comparisons among the groups. Significance was set at a 5% probability level for all observed distinctions. Pearson’s correlation analysis was employed to assess the relationships among serum calcium levels, bone biomarkers, and calcium transporters. ANOVA was chosen due to its suitability for comparing means across multiple groups, followed by Tukey’s post hoc test to identify specific differences between pairs of groups. Statistical analysis and figure generation were conducted using SPSS version 22 for Windows. All measurements were carried out in duplicate, and the data were presented as mean values along with their corresponding standard deviations. It was calculated that a sample size of 8 rats in each group would yield 80% power of detecting statistical significance at the 0.05 α level.

## 3. Results

Table 3 displays the nutritional content of diets used in phase 1 to induce a calcium deficit and phase 2 for administering modified diets to the rats. When compared with the standard diet with a calcium deficit, no significant differences were observed in the nutritional compositions of the modified diets (S, O, OB, OD, and OT), including dry matter, organic matter, protein, fiber, and carbohydrates. However, a modified diet with tempeh (OT) showed a significantly higher nutritional content, specifically in protein, fat, energy, and calcium.

### 3.1. Body Mass Gain and Body Composition

Due to the common association of menopause with increases in body mass and alterations in body composition, we systematically analyzed these parameters in our study. Table 4 presents the outcomes related to body mass gain and body composition in rats throughout the study. Significant increases in final body mass were observed in all ovariectomized groups (O, OB, OD, and OT) compared to the sham group (S). During the calcium deficit period (Phase 1), there was no statistically significant increase in body mass gain in the ovariectomized groups fed with modified diets (OB, OD, and OT) when compared to the S group. However, it is noteworthy that a significant increase in body mass gain during the intervention-modified diet period (Phase 2) was observed between the S and O groups. Furthermore, two distinct scenarios emerged during the second phase. First, the OB group exhibited a body loss of 45% compared to the O group. Second, the OD and OT groups demonstrated body losses of 31% and 34%, respectively, compared to the O group.

Furthermore, these findings illustrate that ovariectomy led to an increase in fat mass. It is noteworthy that, although no significant differences were observed among the various ovariectomized groups, the OT group displayed reduced levels of body fat mass compared to the other ovariectomized groups (O, OB, and OD).

In addition to these findings, Table 4 provides insights into the results concerning food intake, food efficiency ratio (FER), and calcium intake in rats subjected to modified diets during the intervention stage. The FER was significantly higher in the O group compared to the S group, while no differences were noted between the OB, OD, and OT groups when compared to the O and S groups. Furthermore, in contrast to the S and O groups, the OD group exhibited a significantly lower calcium intake, while the OT group demonstrated a significantly higher calcium intake.

### 3.2. Impact on Blood Morphology

Given the typical association of menopause with changes in blood morphology, we analyzed these parameters in our study to discern the alterations. Table 5 provides insights into blood morphology among rats exposed to modified diets. In the comparison between the S and O groups, ovariectomy resulted in a significant increase in leukocytes, neutrophils, lymphocytes, and cholesterol by 86%, 83%, 96%, and 31%, respectively. The results indicated a substantial elevation in the concentration of leukocytes, neutrophils, lymphocytes, and cholesterol following ovariectomy. However, the inclusion of modified diets (OB, OD, and OT) did not induce significant alterations in the levels of these parameters. The OD group demonstrated a notable increase in the levels of monocytes and eosinophils compared to group S, whereas the OB group exhibited significantly higher eosinophil levels than group S. Interestingly, the OT group displayed lower levels of neutrophils and cholesterol among ovariectomized rat groups.

### 3.3. Impact on Calcium in Serum, Calcium in Bone, Calcium in Fecal, and Bone Metabolism Biomarkers

In our study, we performed an analysis of calcium levels in serum, bone, and calcium fecal as well as bone metabolism biomarkers in light of the known correlation between menopause and changes in bone metabolism and calcium status. Table 6 presents the outcomes concerning serum calcium, fecal calcium, and biomarkers associated with bone resorption (pyridinoline, deoxypyridinoline, and C-telopeptide of type I collagen) and bone formation (bone alkaline phosphatase, osteocalcin, and procollagen type I N-terminal propeptide) in rats receiving modified diets.

In the comparison between the S and O groups, ovariectomy led to a slight decrease in calcium content in serum, femur, and feces, although this decline did not reach statistical significance. Notably, within the ovariectomized rat groups, the OB group displayed a significant increase in calcium content in femur bones compared to the O group. Furthermore, despite the significant reduction in serum calcium content observed in the OD group compared to the S group and the OT group compared to the O group, the OD and OT groups exhibited an increased calcium content in feces compared to the other ovariectomized groups.

Additionally, in the comparison between the S and O groups, ovariectomy did not induce alterations in pyridinoline, C-telopeptide of type I collagen, bone alkaline phosphatase, and procollagen type I N-terminal propeptide levels. Intriguingly, within the ovariectomized rat groups, the OT group exhibited a significant elevation in pyridinoline, C-telopeptide of type I collagen, bone alkaline phosphatase, and procollagen type I N-terminal propeptide levels compared to both the S and O groups.

### 3.4. Impact on Histopathological Changes in Femoral Bone

Our study analyzed bone histopathology in order to identify any changes or abnormalities in bone structure, considering the frequent incidence of such changes during menopause. Table 7 details the histopathological alterations observed in various formations of the femoral bone following a 6-week intervention, and the histopathological changes are depicted in Figure 2. The observations reveal that the O group displayed an increased presence of medullary spaces, with a larger surface area occupied by adipocytes and a corresponding area devoid of any bone marrow components, as compared to the S group. This image of the O group indicates an osteoporotic condition. Conversely, the OD and OT groups demonstrated effects comparable to the OB group, illustrating a reduction in the surface area occupied by adipocytes within the femoral bone structure compared to the O group.

### 3.5. Impact on Calcium Transporters

Our study examined calcium transporter expression in ovariectomized rats to understand calcium homeostasis in menopause, which is associated with altered calcium uptake. Table 8 details the mRNA expression of calcium transporters (TRPV5 and TRPV6) in the duodenum and jejunum. Although no statistically significant differences were observed in all groups, the O group showed a slight reduction in TRPV5 and TRPV6 mRNA expression levels in the jejunum and duodenum compared to the S group. Importantly, in both the jejunum and duodenum, the OD and OT groups exhibited elevated TRPV5 and TRPV6 mRNA expression levels when compared to the O group.

### 3.6. Correlation between Calcium Status, Calcium Transporters, and Bone Metabolism Biomarkers

Table 9 provides results from Pearson’s correlation analysis, exploring the associations among calcium status, calcium transporters, selected blood morphology parameters, and bone metabolism biomarkers. The analysis unveiled a positive correlation (*r* = 0.333) between calcium levels in serum and TRPV5 in the jejunum, signifying a positive connection between these variables. Conversely, the most significant negative correlation (*r* = −0.544) was observed between serum calcium and procollagen type I N-terminal propeptide, indicating an inversely proportional relationship.

Moreover, our study unveiled negative correlations between serum calcium levels and biomarkers linked to bone resorption and bone formation. Furthermore, the calcium transporter TRPV6 in the jejunum displayed negative correlations with specific markers representing both bone resorption and bone formation, as outlined in Table 9. These findings emphasize the interconnected relationship between calcium distribution, intestinal calcium transporters, and bone metabolism.

## 4. Discussion

In our meticulously designed experiment, a rat group underwent a sham condition, entailing the administration of a simulated treatment or procedure to account for nonspecific effects. This sham group played a pivotal role as a stringent control, aiding in the differentiation between the authentic effects of the experimental intervention and any potential confounding factors or placebo responses. Simultaneously, we utilized ovariectomized rats, a model involving the surgical removal of ovaries to induce hormonal depletion. This approach facilitated a focused exploration into the precise influences of hormonal fluctuations on key parameters, including blood morphology, calcium status, calcium transport, and bone metabolism.

In comparison to the sham (S) group, our investigation unveiled significant alterations in various body metabolism parameters during menopause. The ovariectomy (O) group exhibited increased body mass gain, fat mass, white blood cells, and cholesterol levels (Table 4 and Table 5). Prior research has extensively documented changes in body composition, weight, and lipid profiles during menopause [41,42,43]. The menopausal transition is intricately linked to heightened adiposity, particularly in the abdominal region. Hypoestrogenism and an imbalanced androgen/estrogen ratio are prominent factors explaining this phenomenon, although other hormonal influences likely contribute [44]. Furthermore, menopause accelerates the process of biological aging [45], reflected in blood white cell composition, providing an indicator of inflammatory and immune status [46]. Menopause is associated with an increase in systemic inflammation and a reduction in T-cell levels. The interplay of heightened fat mass and reduced hormone levels may contribute to inflammation postmenopause [47]. Changes in total cholesterol are independently influenced by menopausal status and are linked to amino acids such as glutamine, tyrosine, isoleucine, and atherogenic lipoproteins [41,48]. From a mechanistic standpoint, the metabolism of glutamine and glutamic acid is proposed to play a role in regulating glucose metabolism and insulin secretion. Additionally, there was a noted increase in leucine concentration, and the hormonal shift during menopause was correlated with elevated tyrosine levels. The elevation in aromatic amino acids and branched-chain amino acid concentrations aligns with insulin resistance. Collectively, these findings suggest intricate connections between lipid profiles, amino acid metabolism, hormonal changes, and insulin-related processes, contributing to a comprehensive understanding of the mechanisms underlying metabolic shifts during menopausal transitions [48,49,50,51].

Furthermore, the ovariectomy condition is observed to diminish calcium status, calcium transporters, and bone formation (Table 6 and Table 8), ultimately resulting in osteoporosis (Figure 2). Notably, in the pursuit of mitigating the effects of ovariectomy, our study represents a pioneering effort to illustrate the significant influence of daidzein and tempeh on calcium status, calcium transporters, as well as bone metabolism and structure in a menopausal osteoporotic animal model. These findings underscore the potential of dietary interventions utilizing daidzein and tempeh to offer novel therapeutic strategies for managing menopausal osteoporosis, warranting further investigation in both preclinical and clinical settings.

In light of the findings presented, it is essential to present the implications of these results within the broader context of bone health, especially concerning menopausal osteoporosis. The observed effects of tempeh and daidzein intake on serum calcium levels, bone biomarkers, and calcium transporters provide valuable insights into potential dietary interventions for managing menopausal osteoporosis. Specifically, the improvements in these parameters suggest that tempeh and daidzein intake may offer promising alternatives to conventional bisphosphonate drugs in promoting bone health and preventing fractures in menopausal women. Additionally, the correlation analyses conducted shed light on the relationships between serum calcium levels, bone biomarkers, and calcium transporters, further emphasizing the complex interplay of factors influencing bone metabolism in the context of menopausal osteoporosis. These below subsections exhibit specifically the results of our current study.

### 4.1. Isoflavone Products and Body Composition in Postmenopausal Osteoporotic

Our findings indicate that the inclusion of pure daidzein and tempeh in the diet for a 6-week duration among ovariectomized rats resulted in a reduction in body mass gain compared to an ovariectomized group receiving a standard diet. Additionally, our investigation revealed that tempeh intake in ovariectomized rats led to a decrease in body fat mass when contrasted with the ovariectomized group subjected to the standard diet, as shown in Table 4. This reduction in body fat mass may have practical implications for bone health, as excess body fat is known to negatively impact bone density and increase the risk of osteoporosis. By promoting a reduction in body fat mass, tempeh intake could potentially contribute to improved bone health outcomes, including enhanced bone density and reduced fracture risk, particularly in postmenopausal women susceptible to osteoporosis. These results underscore the potential of tempeh consumption to positively influence both body mass regulation and composition during the menopausal period.

Similarly, existing literature supports our findings, suggesting that the consumption of tempeh may lead to favorable changes in body weight and adiposity. For instance, studies by Watanabe et al. [52] and Ali et al. [53] observed comparable trends in improvements in body composition associated with tempeh intake, further underscoring the consistent positive influence of tempeh on obesity treatment.

Within the groups of ovariectomized rats, a subgroup receiving tempeh exhibited the lowest levels of neutrophils, along with reduced levels of aspartate aminotransferase, cholesterol, glucose, and triglycerides, as outlined in Table 5. These results suggest that the consumption of tempeh may have favorable effects on immune response and lipid metabolism. Consistent with our findings, previous studies have also indicated that tempeh consumption is associated with enhanced immune parameters and improved lipid profiles. Notably, the works of Suarsana et al. [6] and Afifah et al. [54] offer additional insights into the specific mechanisms underlying these observed benefits, shedding light on the potential pathways through which tempeh positively modulates immune and metabolic parameters. While our findings align with previous studies indicating similar benefits of tempeh consumption on immune parameters and lipid profiles, it is important to consider potential mechanisms underlying these effects. Variability in experimental methodologies, such as differences in animal models, tempeh compositions, and duration of intervention, may contribute to inconsistencies in outcomes across studies.

### 4.2. Isoflavone Products and Calcium Status in Postmenopausal Osteoporotic

Our findings suggest that including pure daidzein and tempeh in the diet for a 6-week duration among ovariectomized rats resulted in decreased serum calcium levels and increased calcium content in the femur compared to the ovariectomized group receiving a standard diet, as depicted in Table 6. Increased calcium levels in the femoral bone due to tempeh or daidzein intake can have significant practical implications for bone health. Higher levels of calcium in the bone contribute to improved bone mineral density and strength, which are crucial factors in reducing the risk of fractures and maintaining overall skeletal integrity. This practical outcome suggests that incorporating tempeh or daidzein into the diet could potentially enhance bone health and reduce the likelihood of osteoporosis-related complications. Remarkably, the observed effects on femoral calcium levels were comparable to the impact of bisphosphonate treatment. These results underscore the potential of isoflavone products, including pure daidzein and tempeh, to positively influence calcium homeostasis during the menopausal period. It appears that isoflavone products may contribute to restoring calcium balance in the menopausal condition across serum, femoral bone, and fecal matters.

This study highlights the potential pathways through which soy isoflavones modulate calcium homeostasis in the context of menopausal osteoporosis. Our current investigation provides compelling evidence suggesting that isoflavones, particularly the abundance of daidzein found in fermented soy foods [28], contribute to restoring calcium balance during the menopausal condition. Through their estrogenic activity, these isoflavones may act as partial substitutes for declining endogenous estrogen levels, potentially regulating bone turnover and calcium homeostasis [55,56]. Furthermore, the observed enhancement in intestinal calcium absorption associated with isoflavone consumption could play a pivotal role in countering the diminished calcium absorption commonly encountered during menopause [57]. These multifaceted effects collectively underscore the potential of isoflavones to positively influence calcium metabolism. Moreover, tempeh serves as a notable source of dietary calcium, surpassing the calcium content found in the standard diets utilized in this study and raw soybeans [58].

### 4.3. Isoflavone Products and Calcium Transporters in Postmenopausal Osteoporotic

In contrast to our prior discovery, which demonstrated that isoflavone products improved TRPV6 levels in the duodenum and reduced TRPV5 levels in the jejunum in healthy female rats [27], our current investigation reveals an intriguing correlation. In this study, tempeh exhibited a high calcium content, contributing to increased calcium intake in the OT group. Despite no significant changes in fecal calcium levels, the OT group displayed potentially elevated calcium absorption. Our findings indicate heightened activity of epithelial calcium transporter channels, TRPV5 and TRPV6, in both the jejunum and duodenum compared to other ovariectomized rat groups (Table 8). This increase in calcium transporter activity suggests a potential improvement in intestinal calcium absorption, which is crucial for maintaining calcium homeostasis and bone health. These changes in calcium transporter activity may have practical implications for bone health, potentially reducing the risk of osteoporosis and related fractures. This suggests that the consumption of isoflavone products, specifically tempeh, may contribute to the restoration of calcium balance during menopause. These results underscore the potential of isoflavones to positively influence calcium absorption and transporter regulation in the gastrointestinal tract. This phenomenon can be explained by the beneficial effects of soy isoflavones on intestinal function, including improvements in secretory capacity, enhancements in the integrity of the intestinal epithelial barrier through the upregulation of tight junction proteins, modulation of intestinal immune or inflammatory responses, and attenuation of histomorphological damage [59].

### 4.4. Isoflavone Products and Bone Metabolism in Postmenopausal Osteoporotic

An intriguing revelation in our current study is that incorporating tempeh into the diet for a 6-week duration among ovariectomized rats led to elevated levels of all serum bone metabolism biomarkers compared to both sham and ovariectomized groups, as illustrated in Table 6. Tempeh intake indicates enhanced bone formation and suggests heightened bone resorption. The observed alterations in bone turnover markers may contribute to improved bone density and strength. Consequently, these findings suggest that tempeh intake could have practical implications for enhancing bone health and reducing the risk of osteoporosis-related fractures. These findings underscore the potential of tempeh to positively impact both bone absorption and formation during the menopausal period, suggesting that tempeh products may play a role in restoring bone metabolism in the context of menopausal conditions.

Our study observed a significant increase of calcium levels in femoral bone following intervention, aligning with previous research indicating the potential of daidzein and tempeh to positively influence calcium metabolism [60]. Similarly, the alterations in bone biomarkers, such as increased bone alkaline phosphatase [61] and procollagen type I N-terminal propeptide [62] levels, are consistent with studies highlighting the bone-forming properties of these phytoestrogens. Although our research did not show the expected impact on TRPV5 and TRPV6 mRNA expression, but these results confirmed several studies [63,64]. Thus, our results contribute valuable insights into the complex mechanisms underlying the bone-protective effects of daidzein and tempeh, emphasizing the need for further investigation to elucidate their full therapeutic potential in managing bone health, particularly in menopausal osteoporosis.

The novel insight into tempeh’s influence on serum bone resorption and formation levels is supported by histopathology results depicting bone microstructure, as shown in Figure 2. The ovariectomized group fed a standard diet exhibited an enlarged surface area dominated by adipocytes, with regions devoid of any bone marrow components. Such an increase in bone marrow adipocytes is known to impede bone formation and fracture healing [65,66]. Furthermore, this enlargement is correlated with endosteum resorption. The endosteum, comprising flattened osteoprogenitor cells and collagenous fibers, plays a crucial role in bone growth and development [67,68,69]. In contrast, the addition of tempeh appears to ameliorate this condition, resulting in a narrower area occupied by bone marrow adipocytes compared to the ovariectomized group fed only a standard diet. This significant finding suggests that a tempeh-rich diet may contribute to the improvement of osteoporotic bone fractures.

It is imperative to underscore the pivotal role of estrogen pathways in bone metabolism [70], given their significance in the pathophysiology of osteoporosis and the rationale behind the use of isoflavones in its prevention and therapy. Isoflavones are known to exert their effects by interacting with estrogen receptors and modulating estrogenic pathways. Specifically, they act as selective estrogen receptor modulators, exhibiting both estrogenic and antiestrogenic properties depending on the tissue and cellular context [71,72].

Regarding their mechanisms of action in bone metabolism, isoflavones can effectively replace endogenous estrogens in metabolic pathways crucial for bone health. One such pathway is the RANKL/RANK/OPG system, which plays a central role in regulating osteoclast differentiation and bone resorption [73]. Additionally, isoflavones enhance the production of osteoprotegerin (OPG), a decoy receptor that binds to RANKL and prevents its interaction with RANK, thus further inhibiting osteoclast formation and activity [74].

By targeting these estrogenic pathways, isoflavones contribute to the enhancement of bone metabolism and a concurrent decrease in bone resorption, ultimately promoting bone health. Furthermore, their ability to modulate other pathways involved in calcium homeostasis [56], such as vitamin D metabolism [75] and calcium transport [64], further underscores their potential as therapeutic agents for osteoporosis.

Furthermore, our study focused on tempeh, which undergoes soybean fermentation inoculated by *R. oligosporus*, leading to an increase in isoflavone content, particularly in the form of aglycone [76]. It is noteworthy that isoflavones in the aglycone form exhibit greater lipid solubility, which facilitates enhanced absorption by the intestines and represents the most bioactive form [77]. Through these mechanisms, isoflavones contribute to the enhancement of bone metabolism and may offer therapeutic potential in the management of osteoporosis.

### 4.5. Isoflavone Products versus Current Osteoporosis Drug

Bisphosphonates play a central role in the treatment of osteoporosis [78]. They function by reducing the risk of fractures through the suppression of bone resorption and improvement of bone strength. However, the clinical use of bisphosphonates for osteoporosis management presents challenges [79], with reported unforeseen adverse effects such as osteonecrosis of the jaw, atypical femur fractures, atrial fibrillation, and esophageal cancer [80]. The primary action of bisphosphonates is the inhibition of bone resorption. Their inherent affinity for bone tissue, particularly osteoclasts, is attributed to the acidic pH within the resorption lacuna during bone resorption, facilitating intracellular uptake [81]. As chemically stable analogs of inorganic pyrophosphate, bisphosphonates serve as potent inhibitors of calcification [82]. By inhibiting bone resorption, bisphosphonates effectively reduce the efflux of calcium from bone, leading to a brief and minor decrease in serum calcium levels [80].

In our comparison between the addition of isoflavone products (pure daidzein and tempeh) and bisphosphonates, our results reveal no significant differences in body composition, blood morphology profiles, calcium status, calcium transporters, and bone metabolism biomarkers. This finding underscores the potential of isoflavone products as safer alternatives to traditional bisphosphonate therapies for managing menopausal osteoporosis. To provide a deeper understanding of these comparisons, we acknowledge the importance of exploring the molecular basis underlying the differential effects of isoflavone products and bisphosphonates on bone health. While bisphosphonates primarily inhibit bone resorption by targeting osteoclast activity, isoflavone products may exert their effects through modulation of various cellular pathways involved in bone metabolism, such as the RANKL/RANK/OPG system and Wnt signaling pathway [12]. Further elucidating these mechanisms could offer valuable insights into the comparative efficacy and safety profiles of these therapeutic approaches.

Additionally, considering other osteoporosis treatments, such as hormone replacement therapy, selective estrogen receptor modulators, and denosumab, alongside isoflavone products and bisphosphonates, could provide a comprehensive overview of available therapeutic options. Future research comparing the effectiveness, safety, and long-term outcomes of these treatments in diverse patient populations is warranted to inform evidence-based clinical decision-making. Overall, our current study suggests that daidzein and tempeh could serve as viable daily alternatives for preventing menopausal osteoporotic risks and are considered safe for long-term consumption.

### 4.6. Isoflavone Products and Their Correlation with Calcium Status, Calcium Transporters, and Bone Metabolism

Aligned with previous research, our study confirms the association between calcium transport and serum calcium levels [27]. Furthermore, our current investigation reveals a reciprocal correlation between serum calcium levels and both bone resorption and formation, emphasizing their involvement in maintaining overall bone metabolism. Calcium homeostasis is intricately governed by processes related to bone. The interaction between cells involved in bone formation and resorption incorporates calcium signals into their differentiation and activation [83]. The dynamic control of calcium signaling, involving the release of calcium from internal stores and its entry into the extracellular fluid, oversees a range of cellular processes. Particularly in osteoclasts, calcium signals play a crucial role in regulating gene transcription, differentiation, and bone resorption [84].

The plausible mechanism behind this phenomenon is attributed to genistein, an isoflavonoid phytoestrogen found in Leguminosae, which may counteract osteoporosis through anabolic bone metabolism. In vitro investigations demonstrate its ability to stimulate protein synthesis in osteoblastic cells, promoting bone formation. Genistein intervenes in osteoblastic bone resorption by hindering the genesis and differentiation of osteoclast-like cells, inducing apoptosis in mature osteoclasts through the Ca^2+^ signaling pathway. This results in a decrease in osteoblastic bone resorption. Additionally, the modulation of protein kinase and tyrosine phosphatase contributes to the reduction in rat bone osteoclast activity. Isoflavones like genistein and daidzein show potential in mitigating bone loss in ovariectomized rats, serving as a model for osteoporosis [60]. Our current findings suggest a dynamic interplay among calcium distribution, intestinal calcium transporters, and bone metabolism in ovariectomized rats.

### 4.7. Study Strengths, Limitations, Future Perspective

This study encompasses several strengths that bolster the scientific rigor and validity of our findings. Firstly, the inclusion of a sham group in our investigation enables the elimination of the impact of ovariectomy on changes in parameters. Additionally, the comparison with the current drug used in osteoporosis management adds valuable context to our research, providing insights into the potential efficacy of daidzein and tempeh as alternative interventions. This meticulous care enhances the reliability of the collected data. The systematic approach in the assessment of calcium and bone metabolism further fortifies the methodological foundation of our study. The integration of these robust methodologies contributes to a comprehensive understanding of the intricate interactions between hormonal fluctuations and physiological parameters, elucidating the potential effects of daidzein and tempeh in the context of menopausal osteoporosis management.

In examining the impact of daidzein and tempeh intake on bone health parameters, it is imperative to acknowledge the inherent limitations of our study design. While our findings reveal promising trends in calcium metabolism and bone biomarkers, it is essential to interpret them within the context of the chosen animal model and its extrapolation to human populations, particularly in postmenopausal bone metabolism.

The utilization of an animal model, although well-established and widely used for studying menopausal osteoporosis, may not fully replicate the intricacies of bone metabolism observed in humans. Variations in physiology, hormonal regulation, and response to interventions between rodents and humans underscore the need for cautious interpretation of our results in clinical settings. Furthermore, while our study provides valuable insights into the potential bone-protective effects of daidzein and tempeh, the extrapolation of these findings to human populations requires careful consideration of factors such as dosage, bioavailability, and long-term effects.

In light of these limitations, our study underscores the necessity for continued research to elucidate the full therapeutic potential of dietary interventions in managing bone health, particularly in menopausal osteoporosis. Future studies employing diverse models, including human clinical trials, and comprehensive analyses of the molecular mechanisms underlying phytoestrogen action will be essential for advancing our understanding and translation of these findings into clinical practice.

While our current study provides valuable insights into the short-term effects of daidzein and tempeh intake on bone health parameters in an animal model, there are indeed several avenues for future research that warrant exploration. For instance, investigating the long-term effects of these diets on bone health would be of great interest. Longitudinal studies could provide a comprehensive understanding of how sustained daidzein and tempeh consumption impacts bone metabolism and fracture risk over extended periods. Additionally, exploring other bone health markers beyond those examined in this study, such as bone mineral density and microarchitecture, could offer a more comprehensive assessment of bone health outcomes.

In addition, it is crucial to recognize the inherent limitations that could impact the interpretation of our findings, despite the fact that our research offers valuable insights into particular parameters associated with calcium transport, calcium status, and bone metabolism. Notably, it is essential to acknowledge the limitations associated with variations in the composition of the diets used. Specifically, the significant differences observed in protein, fat, carbohydrate, and calcium content in the OT diet introduce the possibility that these nutrients could potentially underlie the effects observed in the OT group. Additionally, a notable shortcoming is the lack of measures for vitamin D and vitamin K, which are essential for bone health. Vitamin D, particularly calcitriol, is involved in calcium regulation by diffusing into cells and forming complexes with vitamin D receptors [85]. The absence of vitamin D measurements in our study limits our understanding of its potential influence on calcium metabolism and bone health.

Similarly, the lack of vitamin K measurements is another notable limitation. Vitamin K, comprising fat-soluble vitamers including phylloquinone (K1) and menaquinones (K2), plays a pivotal role in modulating the expression and synthesis of crucial biomarkers associated with bone metabolism [86]. The absence of vitamin K measurements in our study precludes a comprehensive assessment of its impact on bone health parameters. Additionally, the omission of measurements for calcium deficiency biomarkers after the first stage of the study with calcium deficit diet represents a gap in our understanding, as it could have provided valuable insights into the calcium status of the rats and confirmed their potential calcium deficit condition [87].

To address these limitations and strengthen future studies, we propose including measurements for vitamin D, vitamin K, and calcium deficiency biomarkers in subsequent research. These additional analyses would provide a more comprehensive understanding of the complex interplay between these factors and their influence on bone health outcomes. By addressing these gaps in knowledge, future studies can build upon our findings and contribute to a more robust understanding of dietary interventions for bone health.

In addition to these limitations, it is important to address the absence of quantitative bone histomorphometric measurements in our investigation, a methodology exemplified by Dempster et al. [88] and Behets et al. [89]. These studies have demonstrated the utility of quantitative assessments in providing specific measurements of bone formation and resorption parameters. However, our research design opted for a semi-quantitative and qualitative approach, focusing on bone histopathology, aligning with methodologies employed in prior studies [90,91,92]. This choice was motivated by the aim to visually capture microarchitectural changes and pathological conditions in bone tissue induced by tempeh and daidzein in ovariectomized rats. While quantitative measurements offer precise numerical data, qualitative histopathology provides valuable insights into the overall bone microstructure, potentially identifying subtle alterations that may not be captured solely through quantitative means. Recognizing these limitations underscores the need for future research endeavors to address these unexplored facets and further refine our understanding of the interplay between calcium, vitamin D, and bone metabolism.

Our study has significant implications for public health by addressing menopausal osteoporosis through dietary interventions. As the aging population grows, safe and sustainable strategies for managing bone health are increasingly important. Our findings suggest promising alternatives to pharmaceutical drugs, which may have safety concerns with prolonged use. Furthermore, future research endeavours should focus on investigating the long-term effects of daidzein and tempeh intake on bone health, exploring additional bone health markers, and conducting human studies to validate our findings in animal models. These efforts will contribute to a deeper understanding of the potential therapeutic benefits of dietary interventions in managing osteoporosis and promoting bone health in clinical settings. To translate our findings, future research should explore optimal dosages, durations, and modes of administration for daidzein and tempeh. Investigating synergistic effects with lifestyle modifications like exercise could enhance bone health outcomes. Implementing these dietary strategies may face challenges, including cultural preferences, accessibility to tempeh and isoflavone-based food sources, and individual responses to interventions. Interdisciplinary collaborations involving nutritionists, clinicians, and policymakers will be essential to develop tailored, culturally sensitive recommendations. Overall, further research in this area can lead to holistic, evidence-based approaches for long-term bone health in aging populations.

## 5. Conclusions

In summary, our findings indicate that the daily consumption of daidzein and tempeh may improve and restore calcium status, calcium transport, and bone metabolism in ovariectomized rats. Additionally, isoflavone products exhibit effects comparable to bisphosphonate drugs on calcium status, calcium transport, and bone metabolism in ovariectomized rats.

Future inquiries should involve clinical trials to validate and extrapolate findings from animal models, providing a more thorough understanding of the translational potential and safety considerations linked to incorporating isoflavone products into interventions aimed at preserving or enhancing bone health. Clinical trials involving postmenopausal osteoporotic women can offer valuable insights into the efficacy, safety profile, and potential mechanisms of action of isoflavones in human subjects. This, in turn, contributes to the evidence base for informed dietary recommendations and therapeutic strategies in the domain of bone health.

## Figures and Tables

**Figure 1 nutrients-16-00651-f001:**
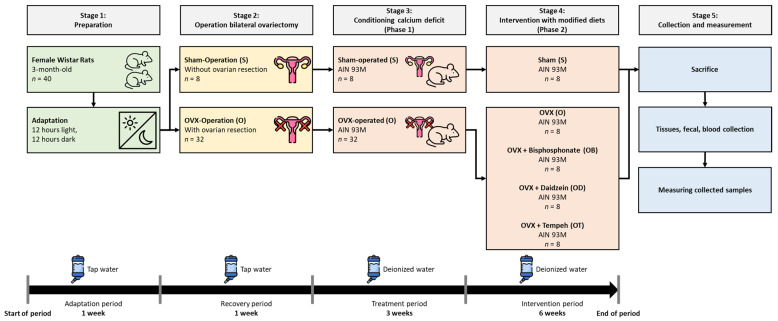
Experimental design of adaptation, treatment, intervention, and measurement periods.

**Figure 2 nutrients-16-00651-f002:**
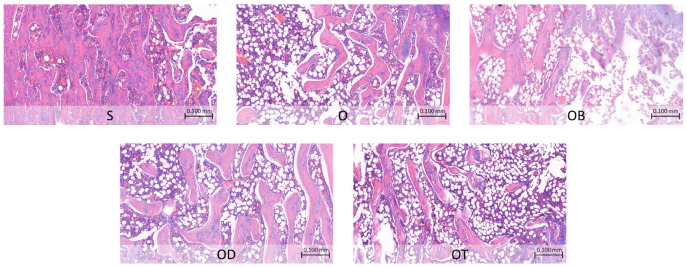
Histopathological changes in femoral bone after a 6-week intervention. S: sham rats fed AIN, serving as the reference group; O: ovariectomized rats fed AIN; OB: ovariectomized rats fed AIN with bisphosphonate; OD: ovariectomized rats fed AIN with daidzein; OT: ovariectomized rats fed AIN with tempeh. Photos were taken under objective 10× and a scale of 0.100 mm.

**Table 1 nutrients-16-00651-t001:** List of formula diets.

Phase	Code	Group	Number of Rats	Diet Formula
Phase 1— Conditioning calcium deficit	AIN	Sham	8	AIN 93M
	AIN_CaDef	OVX	32	AIN 93M with calcium deficit
Phase 2— Intervention modified diets	S	Sham	8	AIN 93M
	O	OVX	8	AIN 93M
	OB	OVX + Bisphosphonate	8	AIN 93M + Bisphosphonate
	OD	OVX + Daidzein	8	AIN 93M + Daidzein
	OT	OVX + Tempeh	8	AIN 93M + Tempeh flour

AIN 93M: a formulated diet designed to provide the necessary nutrients required to maintain rats in this study; AIN_CaDef: an AIN 93M diet without calcium content; OVX: ovariectomized rats.

**Table 2 nutrients-16-00651-t002:** Primer sequences.

Target	Forward Primer	Reverse Primer
Gapdh	TGACTTCAACAGCGACACCCA	CACCCTGTTGCTGTAGCCAAA
TRPV5	CGAGGATTCCAGATGC	GACCATAGCCATTAGCC
TRPV6	GCACCTTCGAGCTGTTCC	CAGTGAGTGTCGCCCATC

Gapdh: Glyceraldehyde-3-phosphate dehydrogenase; TRPV5: Transient Receptor Potential channel family Vanilloid subgroup 5; TRPV6: Transient Receptor Potential channel family Vanilloid subgroup 6.

**Table 3 nutrients-16-00651-t003:** Nutritional content of diets.

Parameter	Type of Diet				
	AIN_CaDef	S & O	OB	OD	OT
	Phase 1	Phase 2	Phase 2	Phase 2	Phase 2
Dry matter (mg/g dry mass)	941.30 ± 1.41 ^ab^	940.15 ± 2.19 ^a^	939.75 ± 0.92 ^a^	938.30 ± 0.14 ^a^	947.45 ± 2.90 ^b^
Organic matter (mg/g dry mass)	939.80 ± 1.34 ^ab^	937.48 ± 2.51 ^ab^	936.82 ± 1.22 ^ab^	935.49 ± 0.04 ^a^	943.95 ± 2.98 ^b^
Protein (mg/g dry mass)	136.04 ± 0.88 ^a^	132.16 ± 0.68 ^a^	133.12 ± 3.59 ^a^	134.23 ± 0.51 ^a^	221.38 ± 1.64 ^b^
Fiber (mg/g dry mass)	40.05 ± 0.69 ^a^	38.61 ± 0.54 ^a^	39.48 ± 1.46 ^a^	46.04 ± 1.65 ^b^	41.69 ± 1.77 ^ab^
Fat (mg/g dry mass)	44.88 ± 1.06 ^b^	43.29 ± 0.10 ^ab^	41.93 ± 0.26 ^ab^	41.41 ± 0.08 ^a^	83.70 ± 1.24 ^c^
Carbohydrates (mg/g dry mass)	826.41 ± 3.20 ^b^	822.85 ± 1.99 ^b^	820.27 ± 3.79 ^b^	816.00 ± 4.12 ^b^	673.66 ± 3.48 ^a^
Energy (Kcal/g)	4333.85 ± 5.39 ^b^	4286.87 ± 3.25 ^a^	4269.86 ± 6.06 ^a^	4265.65 ± 11.87 ^a^	4416.81 ± 0.22 ^c^
Calcium (mg/g dry mass)	0.02 ± 0.01 ^a^	5.06 ± 0.30 ^b^	4.66 ± 0.71 ^b^	4.03 ± 1.12 ^b^	7.06 ± 0.48 ^c^

AIN_CaDef: an AIN 93M diet without calcium content; S & O: sham and ovariectomized rats fed AIN; OB: ovariectomized rats fed AIN with bisphosphonate; OD: ovariectomized rats fed AIN with daidzein; OT: ovariectomized rats fed AIN with tempeh. Phase 1: conditioning calcium deficit; Phase 2: intervention modified diets. Results of ANOVA analysis followed by Tukey’s post hoc honestly significant difference test showing significant differences between types of diet. Data are presented as mean ± standard deviation. ^a, b, c^ represent significantly different mean values ± SD at *p* < 0.05.

**Table 4 nutrients-16-00651-t004:** Final body mass, body mass gain, and fat mass in the rats fed the modified diets during the experimental study.

Parameter	Group				
	S	O	OB	OD	OT
Initial body mass (g)	275.88 ± 18.79	294.50 ± 20.63	292.38 ± 20.32	294.63 ± 20.42	294.50 ± 20.36
Body mass gain in Phase 1 (g)	22.38 ± 13.06	33.00 ± 21.84	39.13 ± 16.50	37.50 ± 5.95	34.63 ± 15.50
Body mass gain in Phase 2 (g)	7.25 ± 7.89 ^a^	27.88 ± 16.98 ^b^	15.25 ± 10.87 ^ab^	19.13 ± 15.80 ^ab^	18.38 ± 7.23 ^ab^
Final body mass (g)	305.50 ± 31.51 ^a^	355.38 ± 23.54 ^b^	346.75 ± 28.41 ^b^	351.25 ± 20.80 ^b^	347.50 ± 33.15 ^b^
Fat mass (g)	62.94 ± 29.63	86.95 ± 14.93	89.53 ± 24.69	88.64 ± 21.43	62.28 ± 22.07
Food intake (g/day)	16.75 ± 1.10	17.78 ± 1.14	17.27 ± 0.96	17.98 ± 1.20	16.80 ± 0.90
FER (%)	43.25 ± 46.96 ^a^	156.88 ± 95.64 ^b^	88.38 ± 63.02 ^ab^	106.25 ± 87.70 ^ab^	109.38 ± 43.07 ^ab^
Calcium intake (mg/day)	85.28 ± 5.60 ^bc^	90.52 ± 5.79 ^c^	80.96 ± 4.52 ^ab^	72.97 ± 4.86 ^a^	119.50 ± 6.42 ^d^

S: sham rats fed AIN, serving as the reference group; O: ovariectomized rats fed AIN; OB: ovariectomized rats fed AIN with bisphosphonate; OD: ovariectomized rats fed AIN with daidzein; OT: ovariectomized rats fed AIN with tempeh. Body mass gain was calculated as the difference in body mass between the end and the beginning of each phase. Phase 1: conditioning calcium deficit; Phase 2: intervention modified diets. FER: Food Efficiency Ratio (%) = weight gain (g)/food intake (g) × 100. Results of ANOVA analysis followed by Tukey’s post hoc honestly significant difference test showing significant differences between types of diet. Data are presented as mean ± standard deviation. ^a, b, c, d^ represent significantly different mean values ± SD at *p* < 0.05.

**Table 5 nutrients-16-00651-t005:** Blood morphology in rats fed modified diets.

Parameter	Group				
	S	O	OB	OD	OT
Erythrocytes (T/L)	7.94 ± 0.40	8.18 ± 0.32	8.09 ± 0.33	8.09 ± 0.51	8.28 ± 0.29
Hemoglobin (g/dL)	14.99 ± 0.65	15.25 ± 0.41	15.44 ± 0.50	15.46 ± 0.59	15.63 ± 0.50
Hematocrit (%)	42.80 ± 1.51	43.55 ± 1.30	43.75 ± 1.87	43.76 ± 2.30	43.80 ± 1.85
MCV (fL)	53.98 ± 1.61	53.40 ± 1.83	54.20 ± 1.24	54.20 ± 1.73	52.99 ± 2.18
MCH (pg)	18.90 ± 0.96	18.69 ± 0.51	19.15 ± 0.67	19.19 ± 0.67	18.88 ± 0.60
MCHC (g/dL)	35.01 ± 1.01	35.03 ± 0.65	35.31 ± 1.10	35.38 ± 0.76	35.65 ± 1.05
Platelets (G/L)	836.25 ± 74.53	838.75 ± 123.39	861.88 ± 144.46	815.63 ± 86.88	852.63 ± 73.45
RDW-CV (%)	12.60 ± 0.50	12.83 ± 0.42	12.83 ± 0.71	12.65 ± 0.58	13.18 ± 0.43
Leukocytes (G/L)	7.34 ± 2.20 ^a^	13.65 ± 2.79 ^b^	13.91 ± 2.10 ^b^	13.07 ± 1.87 ^b^	12.93 ± 1.91 ^b^
Neutrophils (G/L)	0.98 ± 0.28 ^a^	1.79 ± 0.58 ^b^	1.88 ± 0.53 ^b^	1.84 ± 0.55 ^b^	1.64 ± 0.41 ^ab^
Lymphocytes (G/L)	5.45 ± 1.96 ^a^	10.67 ± 2.44 ^b^	10.61 ± 1.82 ^b^	9.56 ± 1.98 ^b^	10.11 ± 1.90 ^b^
Monocytes (G/L)	0.70 ± 0.25 ^a^	0.85 ± 0.32 ^ab^	1.00 ± 0.44 ^ab^	1.26 ± 0.49 ^b^	0.84 ± 0.34 ^ab^
Eosinophils (G/L)	0.20 ± 0.05 ^a^	0.30 ± 0.05 ^ab^	0.38 ± 0.12 ^b^	0.37 ± 0.10 ^b^	0.30 ± 0.00 ^ab^
Basophils %	0.45 ± 0.15	0.29 ± 0.10	0.26 ± 0.09	0.33 ± 0.15	0.31 ± 0.08
ALT (U/L)	37.02 ± 4.96	44.44 ± 8.23	45.79 ± 6.66	54.70 ± 29.77	47.46 ± 7.81
AST (U/L)	155.01 ± 41.32	184.36 ± 40.92	167.86 ± 41.36	251.65 ± 194.95	152.99 ± 45.13
Cholesterol (mg/dL)	81.03 ± 20.85 ^a^	106.51 ± 12.24 ^b^	110.08 ± 16.40 ^b^	115.42 ± 12.75 ^b^	97.35 ± 16.13 ^ab^
Glucose (mg/dL)	115.99 ± 12.96	132.30 ± 15.53	132.59 ± 15.53	135.44 ± 20.36	126.08 ± 10.36
Triglycerides (mg/dL)	210.70 ± 100.72	166.03 ± 71.82	164.48 ± 73.72	163.89 ± 47.69	130.40 ± 34.39

MCV: Mean Corpuscular Volume; MCH: Mean Corpuscular Hemoglobin; MCHC: Mean Corpuscular Hemoglobin Concentration; RDW-CV: Red Cell Distribution Width; ALT: Alanine Transaminase; AST: Aspartate Aminotransferase. S: sham rats fed AIN, serving as the reference group; O: ovariectomized rats fed AIN; OB: ovariectomized rats fed AIN with bisphosphonate; OD: ovariectomized rats fed AIN with daidzein; OT: ovariectomized rats fed AIN with tempeh. Results of ANOVA analysis followed by Tukey’s post hoc honestly significant difference test show significant differences between types of diet. Data are presented as mean ± standard deviation. ^a, b^ represent significantly different mean values ± SD at *p* < 0.05.

**Table 6 nutrients-16-00651-t006:** Calcium in serum, calcium in bone, calcium in fecal, and femoral bone metabolism biomarkers.

Parameter	Group				
	S	O	OB	OD	OT
Calcium in serum (mmol/L)	1.94 ± 0.13 ^c^	1.78 ± 0.07 ^bc^	1.82 ± 0.07 ^c^	1.61 ± 0.18 ^b^	1.42 ± 0.11 ^a^
Calcium in femur (mg/g dry mass)	255.73 ± 63.87 ^ab^	206.98 ± 74.95 ^a^	345.70 ± 41.69 ^c^	331.74 ± 66.53 ^bc^	273.57 ± 54.87 ^abc^
Calcium in faecal (mg/g dry mass)	45.49 ± 9.32	40.14 ± 10.30	38.57 ± 7.65	45.43 ± 4.29	42.20 ± 4.96
Pyridinoline (ng/L)	71.41 ± 14.69 ^a^	80.12 ± 13.19 ^ab^	84.64 ± 9.61 ^ab^	81.15 ± 5.16 ^ab^	91.17 ± 11.73 ^b^
Deoxypyridinoline (ng/mL)	57.99 ± 16.60	59.08 ± 7.25	62.57 ± 8.25	62.63 ± 8.39	68.75 ± 5.44
C-telopeptide of Type I Collagen (ng/mL)	84.06 ± 4.81 ^a^	84.06 ± 4.81 ^a^	87.43 ± 4.81 ^ab^	86.28 ± 9.42 ^a^	97.87 ± 11.55 ^b^
Bone Alkaline Phosphatase (ng/mL)	42.11 ± 4.51 ^a^	42.11 ± 4.51 ^a^	47.34 ± 8.11 ^ab^	41.46 ± 7.55 ^a^	51.37 ± 5.51 ^b^
Osteocalcin (pg/mL)	232.12 ± 60.98	213.33 ± 29.78	235.09 ± 34.95	251.31 ± 40.39	262.52 ± 52.47
Procollagen Type I N-Terminal Propeptide (ng/mL)	8.13 ± 1.30 ^a^	8.52 ± 1.53 ^ab^	8.54 ± 0.80 ^ab^	9.75 ± 1.60 ^ab^	10.46 ± 1.58 ^b^

S: sham rats fed AIN, serving as the reference group; O: ovariectomized rats fed AIN; OB: ovariectomized rats fed AIN with bisphosphonate; OD: ovariectomized rats fed AIN with daidzein; OT: ovariectomized rats fed AIN with tempeh. Results of ANOVA analysis followed by Tukey’s post hoc honestly significant difference test showing significant differences between types of diet. Data are presented as mean ± standard deviation. ^a, b, c^ represent significantly different mean values ± SD at *p* < 0.05.

**Table 7 nutrients-16-00651-t007:** Histopathological changes in several different formation of femoral bone after a 6-week intervention.

Location	Name of Histopathological Change		Group				
			S	O	OB	OD	OT
Methaphyseal trabeculae	Decreased bone	Median	0	1	2	3	2
		Quartile deviation	0	0.25	0	0.5	0
Methaphyseal trabeculae	Trabecular anisotropy	Median	0	0	2	2	1
		Quartile deviation	0.25	0.25	0.25	0.5	0.25
Methaphyseal trabeculae	Trabecular fracture	Median	1	1	1	1	2
		Quartile deviation	0.25	0	0.5	0.5	0.25
Methaphyseal trabeculae	Endosteum resorption	Median	0	0	1	1	2
		Quartile deviation	0.25	0.25	0.25	0	0.25
Bone marrow	Enlargement of the medullary spaces	Median	0	2	3	2	2
		Quartile deviation	0	0.25	0.25	0.25	0
Median sum			1	4	9	9	9

**Table 8 nutrients-16-00651-t008:** TRPV5 and TRPV6 mRNA expression in the duodenum and jejunum.

Parameter	Tissue	Group				
		S	O	OB	OD	OT
TRPV5	Duodenum	1.45 ± 0.60	1.83 ± 0.86	0.95 ± 0.78	0.95 ± 0.68	0.94 ± 0.42
TRPV5	Jejunum	0.68 ± 0.46	0.45 ± 0.36	0.32 ± 0.25	0.56 ± 0.47	0.33 ± 0.10
TRPV6	Duodenum	0.49 ± 0.53	0.27 ± 0.30	0.38 ± 0.31	0.54 ± 1.02	0.31 ± 0.39
TRPV6	Jejunum	2.00 ± 2.94	0.99 ± 1.01	0.52 ± 0.28	0.63 ± 0.39	1.25 ± 1.09

TRPV5: Transient Receptor Potential channel family Vanilloid subgroup 5; TRPV6: Transient Receptor Potential channel family Vanilloid subgroup 6. S: sham rats fed AIN, serving as the reference group; O: ovariectomized rats fed AIN; OB: ovariectomized rats fed AIN with bisphosphonate; OD: ovariectomized rats fed AIN with daidzein; OT: ovariectomized rats fed AIN with tempeh. Values (means ± SD) are expressed as arbitrary units, representing the relative abundance of TRPV5 and TRPV6 proteins compared to the reference protein Gapdh.

**Table 9 nutrients-16-00651-t009:** Pearson’s correlation between calcium status, calcium transporters, and bone metabolism biomarkers.

Correlations		Correlation Coefficient	Significance
Calcium status and calcium transporters	Calcium in femur—TRPV5 in duodenum	−0.322	0.043
	Calcium in serum—TRPV5 in jejunum	0.333	0.047
Calcium status and bone metabolism biomarkers	Calcium in serum—Pyridinoline	−0.358	0.023
	Calcium in serum—Deoxypyridinoline	−0.317	0.046
	Calcium in serum—C-telopeptide of Type I Collagen	−0.363	0.021
	Calcium in serum—Bone Alkaline Phosphatase	−0.362	0.022
	Calcium in serum—Procollagen Type I N-Terminal Propeptide	−0.544	0.000
Calcium transporters and bone metabolism biomarkers	TRPV6 in jejunum—Pyridinoline	−0.374	0.023
	TRPV6 in jejunum—Procollagen Type I N-Terminal Propeptide	−0.339	0.040

TRPV5: Transient Receptor Potential channel family Vanilloid subgroup 5; TRPV6: Transient Receptor Potential channel family Vanilloid subgroup 6. The table depicts statistically significant relationships observed in the correlations between calcium status, calcium transporters, and bone metabolism biomarkers.

## Data Availability

All data generated or analyzed during this study are available from the corresponding author on reasonable request.

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
