# Peer review of "Effect of Tempeh and Daidzein on Calcium Status, Calcium Transporters, and Bone Metabolism Biomarkers in Ovariectomized Rats"

_nutrients, 2024, doi:10.3390/nu16050651_

Round 1

Reviewer 1 Report

Comments and Suggestions for Authors

Abstract:

·         The abstract provides a good overview but could benefit from more precise language regarding the study's scope and findings. For instance, specifying the nature of the "heightened susceptibility to bone fractures" early in the text could provide a better context for the reader.

·         While the abstract mentions "heightened levels of bone turnover markers" and "increased expression" of certain genes in the OT group compared to controls, providing specific data or statistical significance levels could strengthen the impact of these findings. If space permits, including a brief mention of the statistical results would enhance the credibility and specificity of the reported outcomes.

·         The abstract successfully notes that the OT and OD groups exhibited effects similar to the OB group. However, it could further clarify whether these effects were statistically equivalent or only directionally similar. This distinction could help readers understand the potential of daidzein and tempeh as alternatives to bisphosphonate drugs.

·         While the abstract outlines the impact of daidzein and tempeh on various parameters, a brief mention or hypothesis about the underlying biological mechanisms could enrich the reader's understanding. Even a concise statement on whether the effects are thought to be mediated through estrogenic pathways or other mechanisms would be insightful.

·         The abstract does an excellent job presenting the study's findings but could benefit from briefly situating these results within the broader field. A sentence that compares the findings with existing literature or highlights the study's novelty could provide additional context and significance.

1. Introduction

·         The literature review is well-integrated, providing a comprehensive background on the physiological changes during menopause, the role of calcium in bone health, and the potential benefits of isoflavones. It would be beneficial to include a more detailed discussion of previous studies explicitly examining the effects of Tempeh and Daidzein better to position this study within the existing body of research.

·         The introduction effectively explains the rationale for examining Tempeh and Daidzein as alternatives to traditional osteoporosis treatments. However, it could be strengthened by explicitly stating the study's hypothesis or expected outcomes based on the cited literature, thereby providing a more precise direction for the research.

·         While the introduction suggests the study's objectives, explicitly defining the specific research questions or objectives would enhance the reader's understanding of the study's scope and the significance of its potential findings.

·         The introduction briefly mentions using an ovariectomized rat model but does not elaborate on why this model is appropriate for the study's aims. A brief explanation of the choice of this model, including its relevance and advantages for studying menopausal osteoporosis, would be informative.

·         The manuscript introduces bisphosphonates and their drawbacks but does not engage deeply with ongoing debates or controversies in bone health and menopause management. Acknowledging and positioning the study within these debates could enhance its relevance and contribution to the field.

·         The introduction is generally well-written and accessible. However, ensuring that technical terms (e.g., "isoflavones", "TRPV5 and TRPV6 mRNA expression") are clearly defined or explained in lay terms could make the study more accessible to a broader audience, including non-specialists.

2. Materials and Methods

·         While the number of animals used is mentioned, a statistical justification for the sample size could strengthen the study's design, ensuring it has adequate power to detect significant differences.

·         The manuscript could benefit from a clearer explanation of the choice and role of the control groups, particularly how the sham-operated group compares to the ovariectomized groups in terms of expected outcomes.

·         While the diets are described, including information on tempeh production and daidzein dosage, more detail on the nutritional content and how it compares to the rats' needs would be helpful. Specifically, how the chosen diets aim to mimic human consumption levels of these compounds could be clarified.

·         More details on post-operative care and monitoring for complications could provide insight into the welfare of the animals and any potential impact on the study outcomes.

·         The methods for statistical analysis are briefly mentioned. Still, a further elaboration on the choice of tests, especially for the analysis of the data obtained from different groups, could enhance the clarity and robustness of the interpretation of the results.

3. Results

·         The results are well-tabulated, yet the manuscript would benefit from a more nuanced discussion of the implications of these findings in the broader context of bone health, especially in relation to menopausal osteoporosis.

·         The statistical analysis section is briefly mentioned; however, more details on the statistical methods used (beyond ANOVA and Tukey's post hoc tests) and justification for their choice would enhance the reliability of the results.

·         While the manuscript does a good job of comparing the effects of different diets, it falls short in providing a comparative analysis with existing literature, particularly regarding how these findings align or diverge from previous studies on daidzein and tempeh's impact on bone health.

·         The section could benefit from a more critical examination of the limitations of the study design, such as the choice of animal model and its generalizability to human populations, especially considering the complex nature of bone metabolism in postmenopausal women.

·         The manuscript reports statistical significance but does not always translate these findings into biological relevance. Discussing how these changes in biomarkers and calcium levels might impact bone health practically would be beneficial.

·         The results section could be strengthened by suggestions for future research directions, including investigating the long-term effects of these diets on bone health, exploring other bone health markers, and potential human studies.

·         Some data are presented without adequately explaining the units or reference ranges, which could confuse readers unfamiliar with the specific biomarkers or metrics used.

·         Inconsistencies in reporting (e.g., variations in the presentation of statistical data across tables) should be addressed for uniformity.

4. Discussion

·         The discussion begins with a clear summary of the experimental approach and its significance, which is commendable. However, the transition between subsections could be smoother to guide the reader through the narrative more effectively. Explicit connections between the findings and their implications for future research and clinical practice could be emphasized more strongly.

·         The manuscript excellently cites relevant literature to support its findings. However, it would benefit from a more critical analysis of how this study extends, contradicts, or confirms previous work. For instance, while the authors mention studies supporting the beneficial effects of tempeh on body composition and bone health, a deeper discussion on any discrepancies or limitations in the existing literature would provide a more nuanced understanding of the study's contributions.

·         The discussion on the potential mechanisms behind the observed effects of isoflavone products is informative but could be expanded. More detailed hypotheses on how these compounds interact with cellular pathways to influence bone metabolism and calcium homeostasis would enrich the discussion. Additionally, exploring the potential synergistic effects of different components within tempeh could offer insights into its therapeutic potential.

·         The authors acknowledge limitations related to the scope of measured parameters and the study design. This transparency is appreciated, but the discussion could further explore how these limitations impact the interpretation of the results. For example, the absence of vitamin D and K measurements is a significant limitation, given their roles in bone health. Discussing how this might affect the study's conclusions or proposing specific future studies to address these gaps would strengthen this section.

·         The discussion on future perspectives and the potential for dietary interventions in managing menopausal osteoporosis is insightful. However, it could be enhanced by proposing specific, actionable research questions or clinical trials that could test the translational potential of these findings. Additionally, discussing potential challenges or considerations in implementing these dietary strategies in diverse populations would provide a more comprehensive view of the path forward.

·         The comparison with bisphosphonates is essential to the discussion, highlighting the potential of isoflavone products as safer alternatives. Expanding on the molecular basis of these comparisons and considering other osteoporosis treatments could offer a broader context for the study's findings.

Comments on the Quality of English Language

The English in the document is generally clear and understandable, with a coherent structure and logical progression of ideas related to the study's objectives, methodologies, and findings. The scientific terminology and expressions are appropriately used, reflecting familiarity with the academic discourse in nutritional science. There are minor grammatical errors and instances where sentence structure could be refined for improved clarity and readability. These issues, however, do not significantly impede understanding.

Author Response

Dear reviewer,

Thank you very much for taking the time to review this manuscript. Please find the detailed responses below and the corrections highlighted in the re-submitted files.

Reviewer 2 Report

Comments and Suggestions for Authors

The authors investigated an important topic - the effect of soy isoflavones on bone loss in a model of post-menopausal osteoporosis.

My primary concern is the design of the feeding study. The composition of the diets is inadequately described and they appear not to be isocaloric (kcals per kg diet should be listed in Table 3). All macronutrient densities and calcium are significantly different in the OT vs. other diets. This renders difficult  attribution of the outcomes to the isoflavone interventions. All diets should be modified to be similar in nutrient composition. 

The methods should include detailed descriptions of how daidzein and tempeh were incorporated into the diets.

Line 363: Food efficiency ratio should be calculated and presented in Table 4.

The reason for feeding a calcium-deficient diet initially needs to be explained.

Author Response

(The authors gave the same response as above.)

Round 2

Reviewer 1 Report

Comments and Suggestions for Authors

All feedback has been addressed.

Author Response

Dear Reviewer,

We appreciate your time and attention to our manuscript titled “Effect of Tempeh and Daidzein on Calcium Status, Calcium Transporters, and Bone Metabolism Biomarkers in Ovariectomized Rats,” submitted to the Nutrients journal. We acknowledge your review and thank you for considering our work. Thank you for your commitment to the peer-review process.

Reviewer 2 Report

Comments and Suggestions for Authors

The authors have responded to my comments and have added appropriate descriptions of methods to the manuscript.

Still, the authors need to emphasize and more clearly describe the limitations of their study in the differences in nutrient content of the diets. Line 393 - include protein. Around lines 857, the authors need to explicitly state that the significant differences in protein, fat, carbohydrate and calcium in the OT diet introduce the possibility that these nutrients and not tempeh underlie the effects of the OT diet.

Author Response

Dear Reviewer,

Thank you very much for taking the time to review this manuscript in the second round. Please find the detailed responses below and the corrections highlighted in the re-submitted files.

We appreciate your time and attention to our manuscript titled “Effect of Tempeh and Daidzein on Calcium Status, Calcium Transporters, and Bone Metabolism Biomarkers in Ovariectomized Rats,” submitted to the Nutrients journal. We acknowledge your review and thank you for considering our work. Thank you for your commitment to the peer-review process.
